# Purification, Characterization and Evaluation of Inhibitory Mechanism of ACE Inhibitory Peptides from Pearl Oyster (*Pinctada fucata martensii*) Meat Protein Hydrolysate

**DOI:** 10.3390/md17080463

**Published:** 2019-08-08

**Authors:** Pengru Liu, Xiongdiao Lan, Muhammad Yaseen, Shanguang Wu, Xuezhen Feng, Liqin Zhou, Jianhua Sun, Anping Liao, Dankui Liao, Lixia Sun

**Affiliations:** 1Guangxi Key Laboratory of Petrochemical Resource Processing and Process Intensification Technology, School of Chemistry and Chemical Engineering, Guangxi University, Nanning 530004, China; 2Guangxi Key Laboratory for Polysaccharide Materials and Modifications, School of Chemistry and Chemical Engineering, Guangxi University for Nationalities, Nanning 530008, China; 3Institute of Chemical Sciences, University of Peshawar, Khyber Pakhtunkhwa 25120, Pakistan; 4Medical College, Guangxi University of Science and Technology, Liuzhou 545006, China

**Keywords:** pearl oyster, angiotensin-I-converting enzyme inhibitory peptide, antihypertension, Lineweaver-Burk plot, molecular docking

## Abstract

Angiotensin-I-converting enzyme (ACE) inhibitory peptides derived from natural products have shown a blood pressure lowering effect with no side effects. In this study, two novel ACE inhibitory peptides (His-Leu-His-Thr, HLHT and Gly-Trp-Ala, GWA) were purified from pearl oyster (*Pinctada fucata martensii*) meat protein hydrolysate with alkaline protease by ultrafiltration, polyethylene glycol methyl ether modified immobilized metal ion affinity medium, and reverse-phase high performance liquid chromatography. Both peptides exhibited high ACE inhibitory activity with IC_50_ values of 458.06 ± 3.24 μM and 109.25 ± 1.45 μM, respectively. Based on the results of a Lineweaver-Burk plot, HLHT and GWA were found to be non-competitive inhibitor and competitive inhibitor respectively, which were confirmed by molecular docking. Furthermore, the pearl oyster meat protein hydrolysate exhibited an effective antihypertensive effect on SD rats. These results conclude that pearl oyster meat protein is a potential resource of ACE inhibitory peptides and the purified peptides, HLHT and GWA, can be exploited as functional food ingredients against hypertension.

## 1. Introduction

Hypertension is a well-established risk factor for many problems like cardiovascular diseases, chronic kidney diseases, arteriosclerosis, and stroke [1]. The number of adults suffering from hypertension is estimated to increase to 1.6 billion worldwide by 2025 [2]. Angiotensin I converting enzyme (ACE) is regarded a crucial enzyme in regulating blood pressure by promoting the conversion of angiotensin I to angiotensin II in the circulatory or endocrine of human renin-angiotensin system [3,4]. In addition, ACE also converts the vasodilator bradykinin into an inactive peptide via kallikrein–kinin systems [5]. Thus, ACE is regarded as a potential target for antihypertensive pharmaceuticals. Chemically synthesized ACE inhibitors, including captopril, enalapril, ramipril, and isinopril, have grim side effects such as cough, rash, nausea, acute renal failure, and proteinuria [6]. On the contrary, ACE inhibitory peptides derived from natural resources have attracted greater interest in the research community due to their safe and benign nature [7]. In this series, many ACE inhibitory peptides derived from marine sources like cyclina sinensis [8], algae protein waste [9], tilapia by-product protein [10], sardine protein [11], cod skin gelatin [12], and sea cucumber [13] have been reported. These findings conclusively regarded marine proteins as promising sources of ACE inhibitory peptides. 

Pearl oyster (*Pinctada fucata martensii*), a kind of marine pearl shellfish, is widely cultured for pearl production in Southern China with a large number of oyster meat (more than 2000 metric tons) [14]. As a rich source of protein (79.1% protein content on dry basis), the pearl oyster meat could act as a potential source of ACE inhibitory peptides for the functional foods against hypertension [15]. ACE inhibitory peptides in pearl oyster meat protein can be released by enzymatic hydrolysis which is widely applied to improve functional and nutritional properties of proteins [16]. However, the separation efficiency has been facing a bottleneck for isolating ACE inhibitory peptides from the hydrolysate. The separation and purification processes of peptides generally involve several steps such as membrane separation, ion exchange chromatography, gel permeation chromatography (GPC), and high-performance liquid chromatography (HPLC), which greatly complicate the process with a concurrent increase in process cost. To tackle these issues, immobilized metal affinity medium (IMAM), as an effective strategy with low cost for the rapid isolation of ACE inhibitory peptides from protein hydrolysate, has been well documented [17,18]. However, IMAM faces a challenge of non-specific adsorption of protein in hydrolysate, which greatly reduces the selectivity and capacity of the process. To avoid non-specific adsorption, polyethylene glycol methyl ether (mPEG) modified IMAM have been reported with promising results [19,20]. Basically, mPEG plays a role in forming a semi-permeable structure which effectively prevents the non-specific adsorption of large proteins and allows small peptides to pass through to reach affinity sites on the surface of IMAM.

Credited to these reports on the application of modified IMAM, in this study, ACE inhibitory peptides in pearl oyster meat protein hydrolysate (POMPH) were purified through ultrafiltration, mPEG modified IMAM (IMAM@mPEG) and reverse-phase HPLC (RP-HPLC). The sequences of purified peptides were identified by matrix assisted laser desorption/ionization time-of-flight/time-of-flight mass spectrometry (MALDI-TOF/TOF-MS) while their inhibition patterns were illustrated by a Lineweaver-Burk plot and molecular docking. Moreover, the antihypertensive potency of POMPH in vivo was evaluated in SD rats.

## 2. Results and Discussion

### 2.1. Separation and Purification of ACE Inhibitory Peptides from POMPH

The pearl oyster meat protein was hydrolyzed by alkaline protease and the obtained POMPH was ultrafiltered. Two fractions of POMPH with molecular weight less than 5 kDa (D-POMPH) and greater than 5 kDa (T-POMPH) were obtained by ultrafiltration and their ACE inhibitory activity was determined as shown in Table 1. ACE inhibitory activity of D-POMPH fraction (55.25 ± 3.24%) was higher than that of T-POMPH fraction (32.91 ± 1.37%) at a protein concentration of 1.5 mg/mL, this result indicated that small peptides obtained from the ultrafiltration process were more effective in inhibiting ACE activity than peptides of large size which is consistent with previous reports [10,21,22]. D-POMPH due to its higher inhibitory activity, was further purified with IMAM@mPEG by incubating the latter in the former. The immobilization of the target peptides was achieved based on the affinity between the metal ions on the surface of lIMAM@mPEG and amino acid residues of the peptides [17,23]. During the adsorption process, peptides able to bound with the metal ions were retained on IMAM@mPEG which was rapidly separated from the mixture by magnet and washed with phosphate buffered saline (PBS, pH 7.5). The immobilized peptides were eluted with 1.5 M NH_4_Cl (containing 0.5 M NaCl). The ACE inhibitory activity of the eluate was determined as 84.14 ± 2.63%, which was much higher than that of D-POMPH at the same protein concentration. This result indicated that IMAM@mPEG can effectively enrich ACE inhibitory peptides in D-POMPH. The D-POMPH and eluate of IMAM@mPEG were analyzed by GPC (Figure 1) using the standard curve shown in Appendix A. From Figure 1, it can be seen that the proportion of macromolecular components (greater than 1 kDa) in eluate of IMAM@mPEG significantly reduced compared to that in D-POMPH, while the components less than 1 kDa increased from 59.80% to 79.32%. These results indicated that IMAM@mPEG can hinder the adsorption of macromolecular proteins and preferentially enrich more active micromolecular peptides, thus realizing high inhibitory activity. Therefore, the proposed IMAM@mPEG in this study combined with the advantage of size exclusion chromatography and IMAM that can block the non-specific adsorption of macromolecular proteins and effectively enrich ACE inhibitory peptides in one separation step, contributes to an improved selectivity and shorting the purification cycle as well. Moreover, the prepared IMAM@mPEG can be rapidly separated by a magnet, which is much more convenient and time-saving than conventional separation technologies. 

The eluate of IMAM@mPEG with higher ACE inhibitory activity was further separated through RP-HPLC as shown in Figure 2a. Six main fractions were obtained and were assayed for ACE inhibitory activity at a protein concentration of 1.24 mg/mL. A significant difference (*p* < 0.05) was observed between these fractions. Among these fractions, F_5_ and F_6_ showed higher ACE inhibitory activity and were thus further separated using a second RP-HPLC run as shown in Figure 2b,c. F_5_ and F_6_ each was fractionated into three fractions. Their ACE inhibitory activity assayed at a protein concentration of 0.54 mg/mL revealed that fractions F_51_ and F_62_ exhibited higher ACE inhibitory activity (64.73 ± 1.32% and 84.71 ± 1.27% respectively). All fractions showed a significant difference (*p* < 0.05) in ACE Inhibitory activity.

### 2.2. Characterization of Purified ACE Inhibitory Peptides

Molecular masses and amino acid sequences of peptides F_51_ and F_62_ were determined by MALDI-TOF-TOF-MS (Figure 3) revealing their respective molecular masses as 507.117 Da and 334.202 Da, their amino acid sequences were identified as His-Leu-His-Thr (HLHT) and Gly-Trp-Ala (GWA) with IC_50_ values of 458.06 ± 3.24 μM and 109.25 ± 1.45 μM, respectively. HLHT and GWA were synthesized (98% purity) and were found to exhibit ACE inhibitory activity in terms of IC_50_ values of 452.23 ± 2.36 μM and 98.92 ± 5.73 μM, respectively, which were greatly similar to those purified from POMPH. The purified peptides HLHT and GWA are novel ACE inhibitory peptides and have not been reported previously.

### 2.3. Activity-structure Relationship of the Isolated ACE Inhibitory Peptides

According to previous reports, ACE inhibitory peptides are usually composed of 2–12 amino acids [24] and those having hydrophobic amino acids in the primary amino acid sequence have good inhibitory activity [25]. In particular, the presence of aliphatic amino acid at C-terminal and aromatic or basic amino acids at the N-terminal can further enhance the inhibitory activity [26,27]. Thus, a low molecular mass and high hydrophobicity of the purified peptide GWA having aliphatic amino acid-Ala at C-terminal may be the reason for its high inhibitory activity, while the N-terminal His of HLHT is a basic amino acid, which may be responsible for its inhibitory activity. Moreover, the inhibitory activity of peptides is closely dependent on their primary structures, particularly the C-terminal and N-terminal amino acids. The purified GWA and HLHT exhibited similar C-terminal and N-terminal in their primary structures to previously reported ACE inhibitory peptides with high ACE inhibitory activity listed in Table 2 [28], which further strengthens the conclusion that their structures played key role in their inhibitory activity.

### 2.4. Evaluation of Inhibition Pattern of Purified ACE Inhibitory Peptides

Lineweaver-Burk plots were applied to analyze the inhibition patterns of the purified peptides. As shown in Figure 4a, with increasing concentration of HLHT, three straight lines intersected at the same point on the X-axis, suggesting that HLHT is a non-competitive inhibitor binding with ACE at non-active site resulting in an inactive complex irrespective of substrate binding [29]. On the contrary, with increasing concentration of GWA, three lines intercepted the same point on Y-axis (Figure 4b), confirming that GWA competes with substrate for active site of ACE, thus revealing competitive inhibition pattern [30]. In previous studies, multiple inhibition patterns of ACE inhibitory peptides including competitive inhibition, noncompetitive inhibition, and mixed-competitive inhibition have been reported [31,32,33,34,35,36]. The ACE inhibitory peptides derived from marine products exhibiting different inhibition patterns are listed in Table 3. Similar to our study, the ACE inhibitory peptides with different inhibition patterns are observed from a single source such as FEDYVPLSCF and VWDPPKFD from Salmon byproduct protein [31], and CRQHTLGHNTQTSIAQ and EVSQGRP from *Stichopus horren* [32].

### 2.5. Molecular Docking

The intermolecular interaction and potential binding sites of HLHT and GWA with ACE were further elaborated by molecular docking. The main interaction residues at the active site of ACE have been divided into three active pockets, including S1 (Ala354, Glu384 and Tyr523), S2 (Gln281, His353, Lys511, His513 and Tyr520), and S1′ (Glu162 residue) [37]. As seen from Figure 5a, HLHT could not interact with the amino acid residues at the ACE active site, and hence exhibited a noncompetitive manner. Meanwhile, HLHT formed two hydrogen bonds with Ala356 residue of ACE (Figure 5b), which may contribute to its ACE inhibitory activity. However, residues Gly and Trp of the GWA interact with ACE at active site (Figure 5c), GWA formed one hydrogen bond with Ala354 while two hydrogen bonds with Glu384 of ACE (Figure 5d), and hence these hydrogen bonds can be counted as the reasons for stronger ACE inhibitory activity of GWA. Moreover, as Glu384 is an important residue of ACE binding with Zn^2+^ [38], thus can contribute to the ACE inhibitory activity of GWA by avoiding the binding of ACE with Zn^2+^. These findings were in accordance with their ACE inhibition patterns determined by Lineweaver-Burk plots, and hence it was further concluded that the variation in bioactivities of the two peptides could be due to their different natures of interaction with residues of ACE.

### 2.6. Antihypertensive Studies of POMPH on SD rats

The antihypertensive efficacy of captopril and POMPH in vivo was investigated in terms of changes in physiological hypertensive parameters including systolic blood pressure (SBP) and diastolic blood pressure (DBP) after intravenous administration to SD rats and the results are compiled in Figure 6. The control group reported ineffective reduction in SBP and DBP during the 45 min after intravenous administration whereas captopril and POMPH injection revealed a significant decrease in SBP as shown in Figure 6a. The maximum decrease in SBP caused by captopril and POMPH was 27.0 mmHg and 16.7 mmHg, respectively after 20 min. However, a further increase in administrative time beyond 20 min led to an increase in SBP by POMPH groups, while that by captopril remained stable. This result implied that captopril has more durable effects on SBP than POMPH due to the more stable conformation and stringent structure of the former than the latter. The DBP of SD rats after intravenous injection of captopril and POMPH significantly changed in comparison with control group as shown in Figure 6b. The DBP of SD rats in captopril and POMPH groups were decreased to 39.5 mmHg and 20.7 mmHg respectively after 20 min of intravenous administration. These results clearly verified effective antihypertensive effect of POMPH on SD rats, though lower than that of captopril. This might be because captopril is a competitive ACE inhibitor with strong inhibitory activity, while POMPH consisted of different peptides and displayed much weaker ACE inhibitory activity. Endorsing to the significant decrease in SBP and DBP after intravenous administration to SD rats, POMPH can be envisioned as a promising natural source of ACE inhibitory peptides.

## 3. Materials and Methods 

### 3.1. Materials and Chemicals

Fresh pearl oyster (*Pinctada fucata martensii*) meat was obtained from pearl culture base in Beihai, Guangxi Province, China. Alkaline protease (≥250 units/mg) was purchased from Pangbo biological engineering Co., Ltd. (Nanning, China). Toluene and 3-aminopropyl triethoxysilane (APTES) were purchased from Kelon (Chengdu, China). NaBH_4_ was purchased from Sinopharm reagent (Shanghai, China). mPEG (Mn 5000), His-His-Leu (HHL) and ACE (≥2.0 units/mg) were purchased from Sigma-Aldrich (Shanghai, China). Peptides HLHT and GWA were obtained from GL Biochem (Shanghai, China). Trifluoroacetic acid (TFA), and HPLC grade methanol and acetonitrile were purchased from Fluka (Buchs, Switzerland). 

### 3.2. Enzymatic Hydrolysis of Pearl Oyster Meat Protein

Pearl oyster meat protein was hydrolyzed according to reported method [15]. Briefly, 7.0 g pearl oyster meat was chopped and homogenized with 200 mL distilled water and was sterilized in boiling water bath. After cooling, the pH of the solution was adjusted to 9.5 with 1.0 mol/L NaOH, and was further hydrolyzed by alkaline protease with an enzyme/substrate mass ratio of 1/50 at 55 °C for 4 h. The reaction was terminated by raising the temperature to 100 °C and maintaining for 10 min, cooling to room temperature, and finally adjusting pH to 7.0. The obtained POMPH was centrifuged at 4 °C at a rotation of 8000 r/min for 20 min. The supernatant was fractionated by ultrafiltration with molecular weight cut-off membrane of 5 kDa. Two fractions, each corresponding to molecular weight below 5 kDa (D-POMPH) and above 5 kDa (T-POMPH), were collected.

### 3.3. Preparation of IMAM@mPEG

Magnetic IMAM was prepared according to a previously reported procedure [39] and modified with amino group [40]. The magnetic IMAM (0.2 g) were dispersed in toluene under sonication, to which APTES (2 mL) was added, and the mixture was stirred at 110 °C for 8 h. The obtained amino-functionalized IMAM (IMAM-NH_2_) was isolated with a magnet and successively washed with ethanol and deionized water, respectively. Then, mPEG was subjected to aldehyde modification according to a previous report [41]. IMAM-NH_2_ (0.2 g) was dissolved in 80 mL mixed solution of ethanol/water (*v*/*v* = 1:1) containing aldehyde modified mPEG (mPEG-CHO, 1 g). This mixture was stirred at room temperature for 24 h and was mixed with NaBH_4_ (0.076 g). The reaction was continued for 24 h after which the product was dialyzed in distilled water for 48 h. The obtained IMAM@mPEG was magnetically separated and dried in a vacuum freeze drier.

### 3.4. Separation and Purification of ACE Inhibitory Peptide from POMPH

The prepared D-POMPH (MW < 5 kDa) was dissolved in 0.1 M PBS with a concentration of 1 mg/mL. The IMAM@mPEG was mixed with D-POMPH at a ratio of 5:1 (*w*/*v*) for 40 min at 35 °C followed by separating IMAM@mPEG through a magnet, which was washed several times with the same buffer until absorbance of the rinsed buffer at 280 nm reached to a baseline. The adsorbed peptides were eluted with 1.5 M NH_4_Cl (containing 0.5 M NaCl) and isolated with the help of a magnet as mentioned above. The molecular weight distribution of D-POMPH and eluate of IMAM@mPEG were analyzed by GPC at a constant mobile phase of PBS (pH 7, 0.01 M) using Shodex Protein KW-802.5 column with a flow rate of 0.5 mL/min for 50 min at 280 nm. The eluate of IMAM@mPEG was further applied to an RP-HPLC column (Zorbax SB-C18, 150 × 5 μm, Agilent, Santa Clara, CA, USA) for separation using two solvents (solvent A as 0.1% (*v*/*v*) TFA in water, and solvent B as 0.1% (*v*/*v*) TFA in acetonitrile). The gradient (0–35 min, 1%–35% of solvent B) at a flow rate of 0.5 mL/min at 30 °C was used and monitored at 280 nm. The fractions with high ACE-inhibitory activity were collected for the second step RP-HPLC with a linear gradient of solvent B from 1–35% in 4.5 min at a flow rate of 0.5 mL/min at 30 °C. These purification procedures were repeated until sufficient samples were collected to perform an assay of the ACE inhibitory activity and sequencing. 

### 3.5. In Vitro Assay of ACE Inhibitory Activity 

ACE inhibitory activity in vitro was measured according to the reported method [42]. The activity was determined for each step of separation, and fraction with the highest ACE inhibitory activity was collected for further purification. The IC_50_ value (the concentration of peptide inhibiting 50% of enzyme activity) was determined by logarithmic regression analysis.

### 3.6. Identification of ACE Inhibitory Peptides by Mass Spectrometry 

Molecular mass and amino acid sequence of the purified peptides were determined using a 4800 Plus MALDI TOF/TOF^TM^ Analyzer (Applied Biosystems, Beverly, MA, USA). During analysis, each sample was desorbed and ionized at 337 nm and operated in positive ion delayed extraction reflector mode. Spectra were recorded over mass/charge (*m*/*z*) range of 100–1500. Mass spectrometry/mass spectrometry (MS-MS) experiments were achieved through collision-induced dissociation while peptide sequencing was performed via manual calculation.

### 3.7. Determination of Inhibition Pattern

The inhibitory patterns of purified peptides HLHT and GWA on ACE were evaluated by incubating various concentrations (1, 2, 3, 4, and 5 mM) of HHL with ACE in the absence and presence of inhibitory peptides. The concentrations of HLHT were set as 0, 295.86, and 591.72 μM while those of GWA were set as 0, 98.62, and 197.24 μM. The purified ACE inhibitory peptide (30 µL) at various concentrations and ACE solution (30 µL) were mixed and pre-incubated at 37 °C for 10 min. Subsequently, HHL (40 µL) was added and incubated for 5 min at 37 °C. The reaction was terminated by adding 1 M HCl (150 µL). The released hippuric acid (HA) was quantified using HPLC at 228 nm. The ACE inhibitory pattern in the presence of the inhibitor was determined using the Lineweaver–Burk plot, where the reciprocal of HHL concentration was used as the independent variable (X-axis) and the reciprocal of production rate of HA as the dependent variable (Y-axis).

### 3.8. Molecular Docking

The mechanism of interaction of the purified ACE inhibitory peptides bonding with ACE and legitimacy of their activity were elaborated via molecular docking using Autodock software package (Vina, San Diego, CA, USA). The three-dimensional crystal structure of human testicular ACE was obtained from the Protein Data Bank (PDB: 1O86), and the peptides structure were drawn and energy-minimized using Chem Office 2015 software (Cambridge Soft Co., Boston, MA, USA). AutoDocktools was employed to prepare both ACE and peptides for docking. Through Autodock Vina, binding free energy was calculated and peptide displaying the lowest binding affinity to protein was chosen as the best conformation. The visualizations of the protein-ligand structure were shown using PyMol molecular graphics system.

### 3.9. Evaluation of Antihypertensive Activity of POMPH in SD Rats

The antihypertensive efficacy of POMPH was evaluated in male SD rats (10 weeks old, 250 ± 30 g body weight, specific pathogen-free, provided by the animal experimental center of Guangxi medical University, China). All rats were cared and fed following the standards for laboratory animals established by People’s Republic of China (GB14925-2001) [43] and animal handling followed the Declaration of Helsinki [44] and the Guiding Principles in the Care and Use of Animals [45]. The rats were exposed to intraperitoneal injection L-NNA (L-nitro-arginine) once a day for four weeks prior to the experiments [46]. The POMPH at 10 mg/kg, dissolved in a vehicle of 0.9% NaCl, were intravenously administered. The concentration of HLHT and GWA in POMPH were 1.8 µg/mg and 1.2 µg/mg, respectively. The efficacy of POMPH on lowering SBP and diastolic blood pressure (DBP) was compared to that of captopril (10 mg/kg). Control rats were administrated with the same volume of saline solution. SBP and DBP were determined by the noninvasive tail cuff method [47] using BP-2010 (Softron Beijing biotechnology Co., Ltd) at 0 min (before) and 5, 10, 20 and 45 min after the administration.

### 3.10. Statistical Analysis 

All the experiments were performed in triplicate and mean values with ± SD (standard deviation) were reported. All data were analyzed by one-way ANOVA, using SPSS 17.0 software (Chicago, IL, USA). A value of *p* < 0.05 was considered statistically significant. 

## 4. Conclusions

In summary, two novel ACE inhibitory peptides (HLHT and GWA) with respective IC_50_ values of 458.06 ± 3.24 μM and 109.25 ± 1.45 μM were purified from POMPH using multi-step purification including ultrafiltration, IMAM@mPEG, and RP-HPLC. HLHT and GWA exhibited non-competitive and competitive inhibition routes respectively, which were further confirmed via molecular docking simulations. The antihypertensive effect of POMPH was practically confirmed by lowering blood pressure in SD rats. This study concludes POMPH as a promising source of ACE inhibitory peptides and HLHT and GWA purified from POMPH could be deemed as potential anti-hypertensive ingredients for functional food.

## Figures and Tables

**Figure 1 marinedrugs-17-00463-f001:**
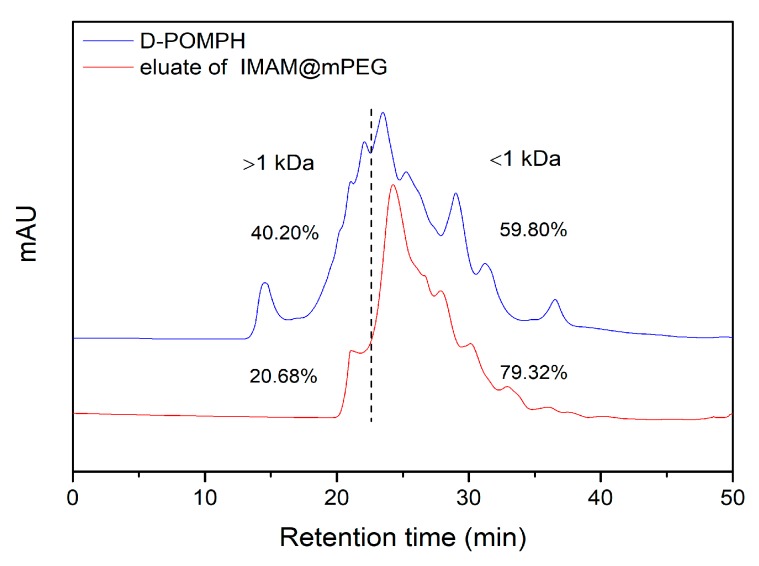
Gel permeation chromatogram of D-POMPH and eluate of IMAM@mPEG.

**Figure 2 marinedrugs-17-00463-f002:**
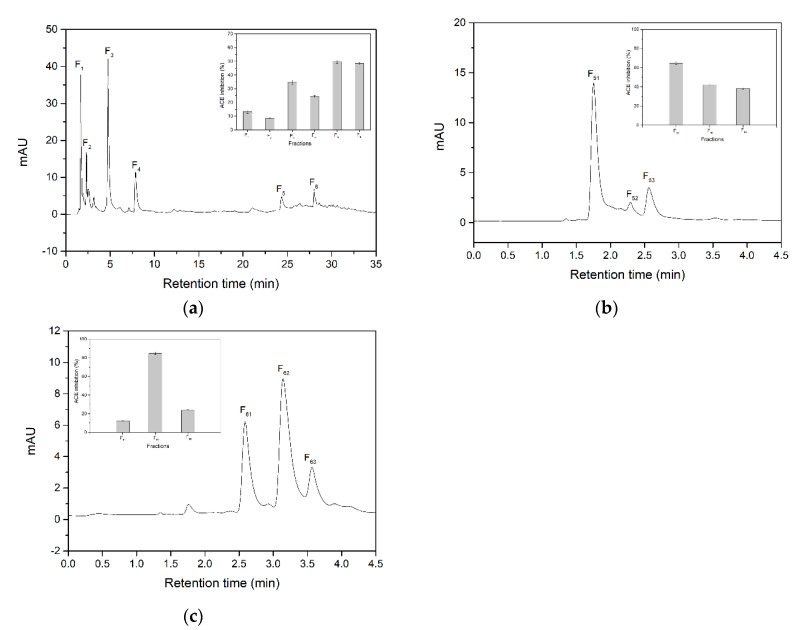
Chromatographic purification and ACE inhibitory activity evaluation of various fractions. RP-HPLC chromatography of eluate of IMAM@mPEG (**a**). ACE inhibitory activity of fractions F_1_ to F_6_ measured at a concentration of 1.24 mg/mL. RP-HPLC chromatogram of F_5_ (**b**) and F_6_ (**c**), ACE inhibitory activity of fractions F_51_ to F_53_ and F_61_ to F_63_ measured at a concentration of 0.54 mg/mL.

**Figure 3 marinedrugs-17-00463-f003:**
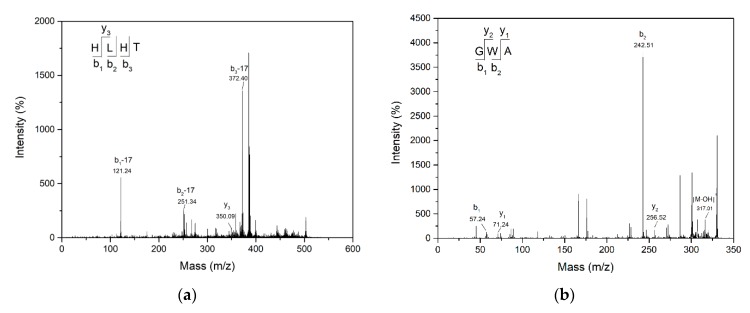
Characterization of molecular mass and amino acid sequence of purified peptides. MS/MS spectrum of molecular ion *m*/*z* 507.117 Da of fraction F_51_ (**a**) and *m*/*z* 334.202 Da of fraction F_62_ (**b**).

**Figure 4 marinedrugs-17-00463-f004:**
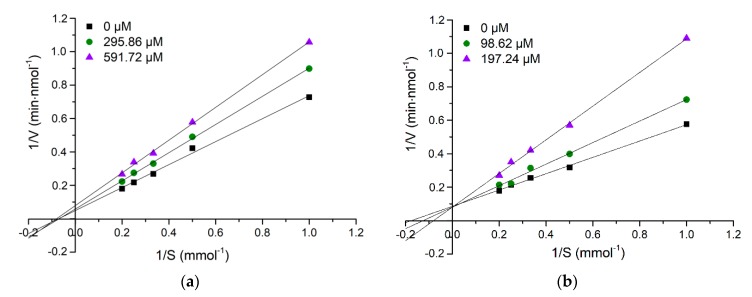
Lineweaver–Burk plots of ACE inhibitory peptides HLHT (**a**) and GWA (**b**) at three concentrations.

**Figure 5 marinedrugs-17-00463-f005:**
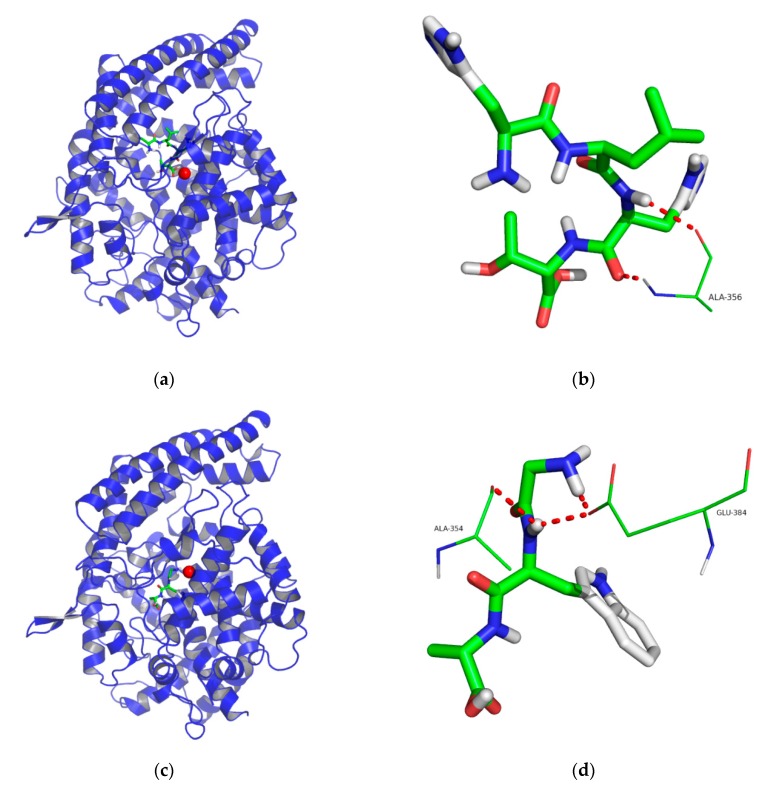
Docking results for the interaction of HLHT and GWA with ACE (PDB: 1O8A). 3D structure of HLHT (**a**) and GWA (**c**) (green) binding with ACE (blue), Zn(II) is represented in red ball. Details of HLHT (**b**) and GWA (**d**) (stick model) interaction at the ACE. Hydrogen bonds are shown with red dotted lines while ACE residues present on binding site are represented as lines.

**Figure 6 marinedrugs-17-00463-f006:**
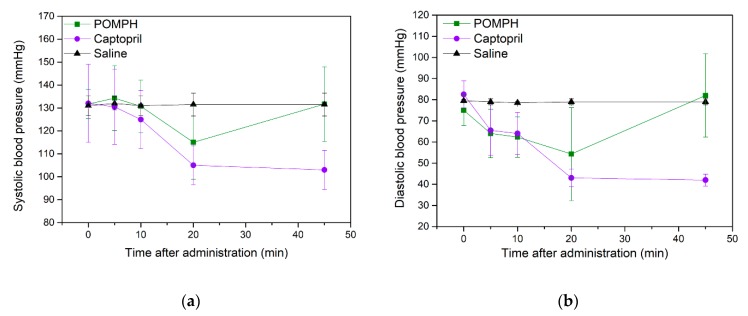
Effect of POMPH on SBP (**a**) and DBP (**b**) of SD rats after intravenous administration.

**Table 1 marinedrugs-17-00463-t001:** ACE inhibitory activity of pearl oyster meat protein hydrolysate and eluate of IMAM@mPEG.

Fraction	T-POMPH	D-POMPH	Eluate of IMAM@mPEG
**ACE Inhibitory Activity (%)**	32.91 ± 1.37	55.25 ± 3.24	84.14 ± 2.63

ACE inhibitory activity of fractions were measured at a concentration of 1.5 mg/mL.

**Table 2 marinedrugs-17-00463-t002:** Summary of ACE inhibitory peptides having similar structure with purified peptides [28].

Amino Sequence	Source	IC_50_ (μM)
VWY**HT**	Izumi Shrimp	28.3
IWH**HT**	Fish (Dried Bonito)	5.1
**HL**PLPLL	Casein	34.4
**HL**PLP	Milk proteins	41
**HL**L	No detected	22.2
**GW**	Soybean	30
**GWA**P	Fish (Sardine muscle)	3.86
A**GW**	Milk derived	<10
I**GW**	Meat protein	<10
**WA**	Fish (Salmon)	277.3

**Table 3 marinedrugs-17-00463-t003:** Inhibition pattern of ACE inhibitory peptides derived from marine products.

Source	Amino Sequence	Inhibition Pattern	IC_50_ (μM)	Reference
Tuna frame protein	GDLGKTTTVSN-WSPPKYKDTP	Non-competitive	11.28	[33]
Cuttlefish (*Sepia officinalis*) muscle protein	VELYP	Non-competitive	5.22	[34]
Oyster protein	VVYPWTTQRF	Non-competitive	66	[35]
Salmon byproduct protein	FEDYVPLSCF	Mixed inhibition	10.77	[31]
	VWDPPKFD	Non-competitive	9.1	
*Stichopus horrens*	CRQHTLGHNT-QTSIAQ	Non-competitive	80	[32]
	EVSQGRP	Mixed inhibition	50	
Freshwater clam (*Corbicula fluminea, Muller*) muscle protein	VKP	Competitive	3.7	[36]
	VKK	Competitive	1045	

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
