# Peer review of "Purification, Characterization and Evaluation of Inhibitory Mechanism of ACE Inhibitory Peptides from Pearl Oyster (Pinctada fucata martensii) Meat Protein Hydrolysate"

_marinedrugs, 2019, doi:10.3390/md17080463_

Round 1
Reviewer 1 Report
In this paper, the authors isolated and determined ACE inhibitory peptides from pearl oyster meat protein hydrolysate (POMPH), and revealed that the POMPH has an effective antihypertensive effect on SD rats. The experiments and the data analysis were adequately performed. The reviewer thinks that this paper is worthy to be published in Marin Drugs after answering the following concerns.
1. P8L205-208
The results of antihypertensive studies of POMPH on SD rats showed that the ACE inhibitory activity of POMP was weaker than that of captopril and the authors think that this difference reflected the difference in the ACE inhibitory activity between captopril and the peptides in POMPH. The reviewer thinks that concentration of peptides, which have the ACE inhibitory activity, e.g., HLHT and GWA, in POMPH is important to work as ACE inhibitors. The reviewer would like to know the concentration of HLHP or GWA in POMPH.
2. P8L211-214
The legend of Figure 6 is the same as that of Figure 5. Please revise it.
3. P10L275
The experimental conditions of “3.7. Determination of inhibition pattern” should be described in more detail.
Author Response
Q1. In this paper, the authors isolated and determined ACE inhibitory peptides from pearl oyster meat protein hydrolysate (POMPH), and revealed that the POMPH has an effective antihypertensive effect on SD rats. The experiments and the data analysis were adequately performed. The reviewer thinks that this paper is worthy to be published in Marine Drugs after answering the following concerns. P8L205-208 The results of antihypertensive studies of POMPH on SD rats showed that the ACE inhibitory activity of POMP was weaker than that of captopril and the authors think that this difference reflected the difference in the ACE inhibitory activity between captopril and the peptides in POMPH. The reviewer thinks that concentration of peptides, which have the ACE inhibitory activity, e.g., HLHT and GWA, in POMPH is important to work as ACE inhibitors. The reviewer would like to know the concentration of HLHT or GWA in POMPH.
Answer: We are thankful to the honorable reviewer for words of appreciation regarding the contents of the submitted manuscript. We understand the reviewer’s concerns about the concentrations of HLHT and GWA in POMPH, which are very important to work as ACE inhibitors. The concentrations of HLHT and GWA in POMPH were determined as 1.8 µg/mg and 1.2 µg/mg, respectively. We have added the data in experimental section of 3.9.
Q2. P8L211-214. The legend of Figure 6 is the same as that of Figure 5. Please revise it.
Answer: We apologize for the mistake and we have revised the legend of Figure 6.
Q3. The experimental conditions of “3.7. Determination of inhibition pattern” should be described in more detail.
Answer: Thanks for your advice. We have added details of experiment in section 3.7 and we hope the revised version will show the experiment process more clearly. The revised part had been highlighted in the manuscript.
Reviewer 2 Report
The authors in this manuscript entitled “Purification, characterization and inhibitory mechanism evaluation of ACE inhibitory peptides from pearl oyster (Pinctada fucata martensii) meat protein hydrolysate” described that pearl oyster meat protein is a potential resource of ACE inhibitory peptides. In my opinion this study is very in tersting but news tests would be needed to evaluate oxidative or pro-inflammatory events and apoptotis in organs and tissues after in vivo treatment with ACE inhibitory peptides from pearl oyster.
Extensive editing of English language and style is required to make the article more understandable.
Author Response
Q1. The authors in this manuscript entitled “Purification, characterization and inhibitory mechanism evaluation of ACE inhibitory peptides from pearl oyster (Pinctada fucata martensii) meat protein hydrolysate” described that pearl oyster meat protein is a potential resource of ACE inhibitory peptides. In my opinion this study is very intersting but news tests would be needed to evaluate oxidative or pro-inflammatory events and apoptotis in organs and tissues after in vivo treatment with ACE inhibitory peptides from pearl oyster.
Extensive editing of English language and style is required to make the article more understandable.
Answer: We are thankful to the reviewer for appreciating our work and providing us with some constructive advice. We believe that the evaluation of oxidative or pro-inflammatory events and apoptotis in organs and tissues can more fully reveal the properties of purified ACE inhibitory peptides and lay an important foundation for their subsequent clinical applications. Available evidences have characterized the crucial role of oxidative stress and inflammation in regulating hypertensive response and the antioxidant and anti-inflammatory activities of ACE inhibitory peptides have been reported previously (Journal of functional foods, 2019, 53: 85-92; Food chemistry, 2017, 221:464-472; Journal of functional foods, 2016, 25: 375-384;). And the apoptotis in organs and tissues were researched for antihypertensive pharmaceuticals such as captopril (Innate immunity, 2017, 23:128-135).
In the present study, we mainly focused on the purification, characterization and inhibitory mechanism of ACE inhibitory peptides. ACE inhibitory peptides are active ingredients of functional foods. By far, there’s no report of apoptotis in organs and tissues about peptides. It provides us a future research direction. Due to the time constraint, we could do the research and investigation in the future study.
We have revised the entire manuscript and have made considerably changes and improvements in terms of English language standards. We hope the revised manuscript meet with the reviewer’s demands.
Round 2
Reviewer 2 Report
Dear Editor,
The manuscript entitled "Purification, characterization and inibitory mechanism
evaluation of AC inhibitoty peptides from pearl oyster (Pinctada fucata martensii)
meat protein hydrolysate"
following the correction can be accepted in present form and published to Marine Drugs.
Best regards